# Studies on Candidate Genes Related to Flowering Time in a Multiparent Population of Maize Derived from Tropical and Temperate Germplasm

**DOI:** 10.3390/plants13071032

**Published:** 2024-04-05

**Authors:** Fengyun Ran, Yizhu Wang, Fuyan Jiang, Xingfu Yin, Yaqi Bi, Ranjan K. Shaw, Xingming Fan

**Affiliations:** 1College of Agronomy and Biotechnology, Yunnan Agricultural University, Kunming 650500, China; ranfenfyun@outlook.com (F.R.); wang4710@foxmail.com (Y.W.); 2Institute of Food Crops, Yunnan Academy of Agricultural Sciences, Kunming 650205, China; jiangfuyansxx@126.com (F.J.); xingfuyin626@163.com (X.Y.); biyq122627@163.com (Y.B.); ranjanshaw@gmail.com (R.K.S.)

**Keywords:** maize flowering time, GWAS, estimated breeding value, tropical maize germplasm

## Abstract

A comprehensive study on maize flowering traits, focusing on the regulation of flowering time and the elucidation of molecular mechanisms underlying the genes controlling flowering, holds the potential to significantly enhance our understanding of the associated regulatory gene network. In this study, three tropical maize inbreds, CML384, CML171, and CML444, were used, along with a temperate maize variety, Shen137, as parental lines to cross with Ye107. The resulting F1s underwent seven consecutive generations of self-pollination through the single-seed descent (SSD) method to develop a multiparent population. To investigate the regulation of maize flowering time-related traits and to identify loci and candidate genes, a genome-wide association study (GWAS) was conducted. GWAS analysis identified 556 SNPs and 12 candidate genes that were significantly associated with flowering time-related traits. Additionally, an analysis of the effect of the estimated breeding values of the subpopulations on flowering time was conducted to further validate the findings of the present study. Collectively, this study offers valuable insights into novel candidate genes, contributing to an improved understanding of maize flowering time-related traits. This information holds practical significance for future maize breeding programs aimed at developing high-yielding hybrids.

## 1. Introduction

Maize (*Zea mays* L.) is the world’s most important food crop and the most extensively cultivated crop. As of 2022, global maize production has surged to approximately 1.2 billion tons, making it the second most important crop worldwide, and the foremost food crop globally (https://www.fao.org/). Notably, China, the second-largest maize producer, contributes significantly to global production, accounting for 24.16% in 2022 alone, with a total production of 277 million tons (https://www.usda.gov/). However, recent years have seen a rise in global temperature by 0.2 °C per decade due to global warming, leading to critical heat stress events that act as a limiting factor for agricultural production, including maize production worldwide [1,2,3]. Maize cultivation has faced varying degrees of challenges related to drought, resulting in a substantial reduction in maize yield ranging from 20 to 30% in China [4,5]. Flowering time is a key trait that determines the local adaptation of plants. The response of maize to drought stress depends on its metabolic capacity, morphological structure, and reproductive period [6]. The flowering stage is a critical window for determining seed sets, and substantial yield loss usually occurs during heat stress [7,8]. Flowering-related traits in maize are considered to be the most critical agronomic characteristics, with a direct or indirect impact on maize yield, disease, stress resistance, and various other aspects. Furthermore, the adaptability of maize varieties to specific environments also plays a significant role in shaping flowering time-related traits.

Exploring the genetic basis of flowering time in maize is crucial for assisting breeders in deciphering the molecular mechanisms underlying flowering genes, thereby fostering a comprehensive understanding of the gene regulatory networks governing traits related to flowering time. Understanding the gene regulatory network of flowering traits is essential for the effective management of maize and the development of high-yielding, resilient maize cultivars. Research conducted by Hall et al. [9] and Dow et al. [10] demonstrated that drought during early anthesis in maize can delay silking, resulting in an extended anthesis silking interval (ASI) and reduced seed yield. Flowering traits in maize are quantitative and regulated by complex genetic mechanisms. Studies on the origin of maize have indicated that flowering time is controlled by numerous minor QTLs located across four major regions [11,12]. Steinhoff et al. [13] investigated the flowering time of 684 individual plants across five maize families, revealing that maize flowering time is influenced not only by major- and minor-effect QTLs but may also involve epistatic effects. Shi et al. [14] performed QTL analysis and identified 17 and 15 QTLs related to anthesis in the RIL and IF2 populations of the maize hybrid Nongda 108, respectively. Interestingly, only one QTL, *qDS1*, was common to both populations, suggesting distinct genetic control mechanisms for anthesis in maize hybrids compared to inbred lines. Previous studies have made notable advancements in identifying candidate genes associated with flowering time and understanding the genetic mechanisms regulating flowering time. In a comprehensive meta-analysis involving 15 studies, Lu et al. [15] identified six QTLs associated with important agronomic traits during maize flowering. Furthermore, through gene ontology (GO) enrichment and cluster analysis, four candidate genes (*ZM00001D005791*, *ZM00001D019045*, *ZM00001D050697*, *ZM00001D011139*) related to flowering were successfully identified. Several other genes associated with maize flowering, including the indistinguishable gene (*id1*) [16], delayed initiation of flowering (*dlf1*) [17,18], vegetative germline transition (*vgt1*) [19], the corresponding control of thermocycle (*zmCCT*) [12,20], dwarfing (*Dwarf 8*), CONSTANS-like (*conz1*) [21], Transposon insertions into the maize gene (*zfl1*) [22], *ZEA CENTRORADIALIS 8 (ZCN8)* [23,24], *ZmPRR37* [25], *ZMGI2* [26], *ZmNF-YC2* [27], *ZMCCT9* [28], *ZMCCT10* [29], and *ZmPHYB2* [30], have also been documented in previous studies. Although progress has been made in identifying candidate genes for maize flowering time, it still lags behind model crops, such as *Arabidopsis thaliana*, in the identification of genes linked to maize anthesis. To date, our understanding of the genetic and molecular regulatory mechanisms associated with maize anthesis remains limited.

SNP markers offer substantial theoretical and practical value [31,32]. SNPs are frequently used in genome-wide association (GWAS) analysis, aiding in deciphering the genetic mechanisms governing the quantitative variation in phenotypic traits controlled by multiple genes in numerous crop species. For instance, Li et al. [33] conducted a GWAS study based on SNP and haplotype analysis, employing 39,350 high-quality SNPs across 410 inbred lines. They successfully identified haplotypes associated with flowering time and photoperiod sensitivity. The haplotypes identified at these loci accounted for 17.5–18.6% and 11.2–15.5% of the phenotypic variations in the two traits, respectively. Li et al. [34] performed a GWAS study involving two NAM populations (CN-NAM, US-NAM) and a natural association panel (Ames), utilizing nearly 1 million SNPs. Through this analysis, they identified approximately 1000 SNPs associated with flowering time, along with 220 candidate genes located within a 1 Mb distance. In another study, Wallace et al. [35] selected 15 WEMA populations for a GWAS focusing on flowering traits in maize under both drought stress and well-watered conditions. They identified 115 SNPs significantly associated with flowering time under well-watered conditions and 108 SNPs significantly associated with flowering time under drought stress conditions. The SNPs identified in these two environments accounted for 80% and 36% of genetic variance, respectively. In addition, a GWAS study by Bezrutczyk et al. [36] revealed a correlation between variability in *ZmSWEET13s* and maize flowering time.

Although certain insights have been gained from studies investigating the genetic mechanisms underlying maize flowering traits, it is important to note that these traits exhibit quantitative characteristics. The diversity in materials and methodologies across various studies on related traits has led to disparate outcomes [37,38]. Hence, employing diverse populations, sequencing technologies, and multiple genetic mapping methods across various environments, regardless of the consistency or variability of their findings, can offer valuable insights into unraveling the genetic basis of maize flowering time-related traits. This improved understanding has the potential to significantly contribute to deciphering the genes that govern these traits in maize. In this study, we crossed four inbred lines, CML382, CML171, CML444, and Shen137 (used as female parents), with Ye107 (used as the male parent). Subsequently, we generated four F_8_ RIL subpopulations through successive selfing and developed a multiparent maize population. The primary objective of this study was to identify candidate genes associated with flowering time-related traits in maize, including flowering period, silking period, and anthesis–silking interval. This objective was accomplished through GWAS analysis. The findings of this study deepen our understanding of the genes and genetic mechanisms governing maize flowering-related traits and provide valuable insights that can contribute to the breeding of flowering-related traits in maize.

## 2. Results

### 2.1. Phenotypic Analysis of Flowering Time-Related Traits

The flowering time-related traits of the multiparent population exhibited substantial variations across the three environments: Yanshan in 2021 and 2022, and Jinghong in 2021 (Table 1). Among the three flowering-related traits, ASI displayed the most significant variation, with a coefficient of variation (CV) ranging from 64.8% to 139.5%. The most substantial fluctuation was observed for ASI at Yanshan (YS21-ASI) in 2021, which was attributed to drought conditions during the mid-flowering stage, leading to delayed silking in some plants. Comparing the results of gene tests in arid environments with those in other environments may help detect gene regulation of ASI. The variation in DTA ranged from 6.0% to 7.6%, whereas the variation in DTS ranged from 6.6% to 7.6% across the three environments. The broad-sense heritability for the three flowering time-related traits was 38.2%, 42.1%, and 11.9% for DTA, DTS, and ASI, respectively (Table 2). Extensive variation and low broad-sense heritability suggest that traits related to flowering time are affected by environmental factors, with ASI being the most prone to such influences.

Correlation analysis was conducted for these three traits across different locations, and the results are presented in Figure 1 (Figure 1). It was observed that ASI was negatively correlated with DTA and positively correlated with DTS, and there was no significant linear relationship between ASI and both DTA and DTS. DTA was positively correlated with DTS, and there was a significant linear relationship between DTA and DTS. Although the traits displayed some degree of correlation, only DTA and DTS exhibited significant correlations under the same environmental conditions. In summary, a strong overall correlation was observed between the flowering and silking period in maize. 

### 2.2. Population Structure Analysis

Population structure analysis using Admixture revealed the presence of four distinct subpopulations within the multiparent population comprising 696 RILs at K = 4. Specifically, 174 RILs were assigned to taxon 1, 168 to taxon 2, 196 to taxon 3, and 158 to taxon 4, based on the obtained genetic structure. As illustrated in Figure 2, when K = 4, the distribution of the four different colored blocks corresponds to four separate groups, each characterized by a pure lineage. Subsequently, PCA using GCTA [39] confirmed the clustering of all 696 RILs into four distinct subgroups (Figure 3), denoted as Pop1, Pop2, Pop3, and Pop4. 

### 2.3. GWAS Analysis

GWAS analysis of the three flowering traits (DTA, DTS, and ASI) in the multiparent population across 2021 in Yanshan, 2022 in Yanshan, and 2021 in Jinghong, along with BLUP values, was conducted using the GEMMA’s MLM model. The calculation was performed using the formula −log10(1/588,416) (where 588,416 is the number of SNPs used in the experiment), resulting in the identification of 556 significant SNPs at a threshold value of *p* < 10^−5^ (Appendix A). 

#### 2.3.1. GWAS Analysis for ASI

GWAS analysis of ASI (Figure 4) showed that 259 SNPs were detected at the threshold value. Among them, 10 were detected in 2022 Yanshan, 18 in 2021 Yanshan, 216 in 2021 Jinghong, and 15 were detected for BLUP values.

#### 2.3.2. GWAS Analysis for DTA

GWAS analysis of DTA (Figure 5) showed that 130 SNPs were detected at the threshold value. Among them, 12 were detected in 2022 Yanshan, 24 in 2021 Yanshan, 61 in 2021 Jinghong, and 33 were detected for the BLUP values.

#### 2.3.3. GWAS Analysis for DTS

GWAS analysis of DTS (Figure 6) showed that 167 SNPs were detected at the threshold value. Among them, 20 were detected in 2022 Yanshan, 19 in 2021 Yanshan, 71 in 2021 Jinghong, and 57 were detected for BLUP values.

Notably, several multi-environment, multi-trait co-localized SNPs were identified within a range of 20 kb upstream and downstream of specific SNPs (Appendix A). These co-localized SNPs were distributed across chromosomes 1, 3, 4, 5, 7, 8, and 10.

### 2.4. Identification and Functional Annotation of Candidate Genes

In this study, candidate genes related to flowering time were identified by screening a 20 kb region upstream and downstream of the significant SNPs identified through GWAS, employing the B73_RefGen_v4 reference genome. Subsequently, based on functional annotation and RNA-Seq expression data (https://maizegdb.org/), 12 candidate genes potentially related to maize flowering time were identified (Table 3). Among them, five candidate genes were associated with both the flowering and silk-spitting periods, while three candidate genes were simultaneously associated with the silk-spitting period and the anthesis and silking interval. A relatively strong correlation was observed between the flowering time and the silk-spitting period.

### 2.5. Haplotype Analysis

*Zm00001d028325* accounted for 5.90% of phenotypic variation. A significant difference was observed between haplotypes Hap1 and Hap2 (Figure 7), with haplotype frequencies of 97 for Hap1 and 342 for Hap2 out of 696 RILs (Table 4). Considering DTA length as the superior haplotype, Hap2 exhibited an average pollen dispersal duration of 87.0 days, which was delayed compared to the haplotype Hap1, with an average pollen dispersal duration of 82.5 days (Table 5).

The gene *Zm00001d028615* accounted for 12.43% of the phenotypic variation and exhibited both Hap1 and Hap2 haplotypes (Figure 8), with a frequency of 213 Hap1 and 125 Hap2 haplotypes out of the 696 RILs (Table 4). The superior haplotype, Hap2, had an average ASI duration of 1.9 days, indicating a shorter ASI than haplotype Hap1, which averaged 2.1 days for ASI duration (Table 5).

The gene *Zm00001d048680* exhibited four haplotypes: Hap1, Hap2, Hap3, and Hap4. The frequency distributions of these haplotypes were as follows: Hap1 (131 of 696 RILs), Hap2 (3), Hap3 (10), and Hap4 (21). In the YS22-DTS environment, a significant difference was observed between haplotypes Hap1 and Hap3 (*p* ≤ 0.05), and the Hap3 haplotype was significantly different from Hap4 (Figure 9) (*p* ≤ 0.001). When a short DTS was used as the superior haplotype, the superior haplotype for *Zm00001d048680* in this environment was Hap3. Specifically, in the YS22 DTA environment, the Hap1 haplotype was significantly different from Hap3 (*p* ≤ 0.05) and the Hap3 haplotype was significantly different from Hap4 (*p* ≤ 0.01) (Figure 10). Conversely, when the long DTA was used as the superior haplotype, the superior haplotype for *Zm00001d048680* in this environment was Hap4. In the JH21 DTA environment, the Hap1 haplotype was significantly different from Hap3 (*p* ≤ 0.01) and the Hap3 haplotype was significantly different from Hap4 (*p* ≤ 0.0001) (Figure 11). Notably, the same gene did not exhibit superior haplotypes for different traits in different environments, and the significant differences between haplotypes varied.

### 2.6. Analysis of Estimated Breeding Value (EBV)

GWAS, originally introduced by Merikangas and Risch [40], enables the use of genome-wide SNPs and phenotypic information from natural or multiparent populations to construct a mathematical model. This model can then be applied to SNP markers in a target population to predict the phenotypic (breeding) values of individual agronomic traits. This forms the foundation of genomic selection (GS) in breeding programs [41]. 

In this study, we used high-density SNP markers spanning the entire genome, along with phenotypic data from the multiparent population, to estimate the breeding values of flowering traits in each subpopulation (Pop1, Pop2, Pop3, and Pop4) (Appendix A). As shown in Table 6 (Table 6), the breeding values of DTA, DTS, and ASI in Pop4 were predominantly positive. Specifically, the breeding values of ASI in Pop4 exhibited a higher number of positive EBVs compared to negative EBVs. In contrast, in Pop1, Pop2, and Pop3, more negative EBVs were observed than positive EBVs for the ASI. This suggests that Pop1, Pop2, and Pop3 may play a role in shortening the ASI. For DTA, Pop2 and Pop4 showed a higher number of SNPs with positive EBVs than negative EBVs, indicating the likelihood of these populations leading to a later DTA. Conversely, only Pop4 had more SNPs with positive EBVs than negative EBVs for the DTS, suggesting a potential for later silking in this population. Pop2 may have had a late DTA and earlier DTS, which may have led to Pop2 having a shorter ASI (ASI = |DTS-DTA|). Pop3 had the largest percentage of negative estimated breeding values for ASI (64.55%) which may have also led to a shorter ASI.

## 3. Discussion

GWAS has emerged as a valuable tool in agricultural research in recent years. Previous GWAS studies have successfully identified candidate genes associated with various traits in maize. Alexandrov et al. [42] reported a significant number of candidate genes related to maize flowering traits. However, their focus was primarily on sequencing these candidate genes without annotating their function in maize. Flowering time is a complex process that encompasses the development of most plant organs and is influenced by various factors including species, geography, environmental conditions, and the extent of evolutionary processes. Therefore, the gene regulatory networks involved in flowering are complex. Prior studies have categorized the maize floral gene regulatory network into four major pathways: photoperiod, autonomous, gibberellin, and age [43]. Extensive research has shed light on the genetic mechanisms governing floral regulation in rice and *Arabidopsis thaliana*. Since maize is a cereal crop similar to rice and *Arabidopsis thaliana* and serves as a model plant, it is possible that the functional genes identified in these crops may have analogous functions for flowering time-related genes in maize.

In the present study, we observed that some candidate genes associated with maize flowering time-related traits were linked to flowering time in different crops. For instance, Wang et al. [44] demonstrated significant changes in gene expression related to cellular rescue, transcription, signal transduction, and cellular transport during pollen germination and pollen tube growth in *Arabidopsis thaliana*. This process led to the emergence of many new transcripts, suggesting the crucial role of these newly expressed genes in this complex process. In our study, the candidate gene *Zm00001d020229*, which is a homolog of AT4G23630 in *Arabidopsis thaliana*, was found to be responsible for shaping and maintaining the curvature of endoplasmic reticulum (ER) membranes. It is implicated in maintaining the structure and function of the cortical ER network [45]. Given the crucial role of the ER in plant growth and development, the impact of RTNLB1 on maize flowering may be indirect, exerted through its influence on ER structure and function. *Zm00001d021470*‘s ortholog in Arabidopsis, GSTF10, regulates the drought stress and abscisic acid signaling pathways. Specifically, GSTF10 functions to mitigate the damage caused by drought by scavenging excess cellular ROS (reactive oxygen species) under drought conditions. It is worth noting that under drought stress, the anthesis and silking interval (ASI) in maize tends to be prolonged [46], resulting in a decrease in maize yield. Therefore, GSTF10 may be involved in the regulation of flowering time-related traits in maize. *AT1G73060.1* (LPA3), the ortholog of *Zm00001d028295* in Arabidopsis, belongs to the LPA3 protein family. LAP3, when mutated, leads to decreased levels of PSII (photosystem II) and impaired photosynthesis [47], thereby affecting plant development and growth. Hence, LPA3 may be associated with the modulation of plant growth and developmental processes, potentially influencing the onset of flowering in maize. *AT3G1320*, an ortholog of *Zm00001d039455* in Arabidopsis, belongs to the PRA1 protein family and is believed to play a role in the signaling process of the phytohormone gibberellin in Arabidopsis [48]. *Zm00001d020229* is likely to have a regulatory role in both the pollen dispersal and the silking period. *Zm00001d047209*, the ortholog of *AT5G09410* in Arabidopsis, may share a similar function with *AT5G09410* in regulating drought response in *Arabidopsis thaliana* [49]. *Zm00001d047209* was found to regulate the ASI, which can be used to evaluate drought tolerance in inbred lines of maize [50]. Therefore, we hypothesized that this gene is involved in the regulation of the anthesis and silking interval. Another candidate gene identified in this study, *Zm00001d033666*, is a homolog of *Os03G0704100* (in rice) and may function similarly to *Os03G0704100* in mitigating salt, cold, and drought stress. *Os03G0704100* is related to the cytokinin signaling pathway [51]. *Zm00001d033666* is co-detected for DTS and ASI. Cytokinins are closely linked to the growth and development of plants and their response to drought. It has a significant impact on the silking stage of maize, potentially delaying the silking time of the female ear [52]. Therefore, we hypothesized that this gene is involved in the regulation of the anthesis and silking interval. Other candidate genes identified in this study, including *Zm00001d039456* and *Zm00001d050697*, have homologs with similar functions in *Arabidopsis thaliana* [15,53] and are involved in the putative regulation of their respective localized floral traits. The consistency in the functions of these candidate genes, aligned with previous studies, strongly suggests the accuracy of the findings of the present study. 

We focused on the candidate genes *Zm00001d028325*, *Zm00001d02615*, and *Zm00001d048680* for haplotype analysis. Different haplotypes of *Zm00001d028325* showed significant differences in their association with DTA, suggesting their potential role in regulating the maize pollen dispersal period. Although different haplotypes of *Zm00001d02615* had insignificant associations with ASIs, its superior haplotype Hap2 showed a shorter ASI, possibly related to drought resistance and indirect regulation of the anthesis and silking interval in maize. The haplotypes of *Zm00001d048680* displayed significant differences in association with DTA and DTS, with varying degrees of significance across different environmental traits. Therefore, this gene is likely to be related to maize pollen dispersal. Notably, *Zm00001d048680* exhibited significant differences in different environments for the same trait (it exhibited different superior haplotypes in different environments for DTA). In the JH21-DTA environment, Hap3 emerged as the superior haplotype, whereas in the YS22-DTA environment, Hap4 emerged as the superior haplotype, indicating the involvement *of Zm00001d048680* in regulating the maize pollen dispersal period. The majority of the selected candidate genes regulate both pollen dispersal and the silking period, thereby regulating multiple traits. The correlation between pollen dispersal and the silking period was very high, whereas the correlation between anthesis and the silking interval, as well as the remaining two traits, was relatively low. When investigating the regulatory mechanisms of pollen dispersal and the silking period in maize, it is possible to explore whether the genes involved in pollen dispersal or the silking period also regulate the other two traits.

In this study, breeding values representing the expected phenotypic values of flowering time-related traits were predicted for each subpopulation (Pop1, Pop2, Pop3, and Pop4) using high-density SNPs spanning the entire genome and phenotypic data from a multiparent population. The ASI breeding values predicted for Pop1, Pop2, and Pop3 were predominantly negative, signifying an overall reduction in ASI length. The data indicated that Pop2 has the potential to shorten the ASI of maize (Table 7), suggesting that this subpopulation may be suitable for drought-resistant breeding in maize. CML384, CML171, and CML444 are tropical maize germplasms, whereas Shen137 is a temperate maize germplasm. In recent years, there has been a growing focus on studying maize ASI in stress-resistant germplasms [54,55,56,57]. Prolonged ASI in maize can result in a reduction in grain number per ear, significantly affecting the overall yield [58,59]. The ideal maize type, as proposed by Mock et al. [60], is characterized by a small ASI. Therefore, studying the genetic mechanisms underlying the ASI trait in maize is essential for effective breeding in maize. Tropical maize germplasms are drought-tolerant in terms of their ASI traits. CML444 consistently maintained a short ASI in almost all environments and contributed favorable alleles in bins 1.04, 5.01, 7.04, and 8.06 [61]. The CIMMYT’s DTMA research project demonstrated that CML444 is a highly drought-tolerant variety with a short ASI (http://dtma.cimmyt.org/). CML171 has been less studied for flowering time-related traits in maize, but it has frequently been used in maize breeding to study hybrid vigor [62,63,64,65]. According to the estimated breeding value (EBV) in this study, Pop2 may have a shorter ASI, and Pop3, obtained using CML171 as a parent, may also have a shorter ASI. This proves that using CML444 and CML171 as parents is feasible for obtaining hybrids with shorter ASIs. Both CML444 and CML171 have significant potential as parental lines in future breeding programs aimed at developing drought-tolerant hybrids.

In summary, this study identified 12 candidate genes through GWAS in a multiparent maize population. Some of these genes have direct homologs that have been reported in the literature to be associated with flowering time. The identification of overlapping loci in both GWAS and QTL mapping indicates a highly reliable association analysis. Among these candidate genes, *Zm00001d048680* showed significant haplotype differences in multiple environments and was associated with biological phosphorylation, a process potentially linked to plant hormone metabolism. The involvement of this gene in biological phosphorylation suggests a potential connection to maize flowering, as hormonal regulation often plays a crucial role in the flowering process. Future research could further analyze the function of this gene to enhance our understanding of the regulatory mechanisms underlying flowering time.

## 4. Materials and Methods

### 4.1. Experimental Materials and Field Trials

This study used an elite inbred line, Ye107, which is widely used as a male parent in maize breeding programs in China. Four inbred lines, CML384, CML444, CML171, and Shen137 (Table 8), were selected as female parents due to their substantial genetic variation in the anthesis and silking interval (ASI). These female parental lines were crossed with Ye107 to produce F1s, and subsequently, four subpopulations of F8 RILs were developed through seven consecutive generations of selfing using the single-seed descent method. The common parent, Ye107, is one of the founding parents of maize in China, belongs to the Reid heterotic group, and holds significance in maize breeding in China. The four subpopulations comprised 696 RILs, with Pop1 consisting of 174 lines (CML384 × Ye107), Pop2 of 168 lines (CML444 × Ye107), Pop3 of 196 lines (CML171 × Ye107), and Pop4 of 158 lines (Shen137 × Ye107). All RILs were planted in Yanshan, Yunnan Province, during the spring of both 2021 and 2022. For this trial, a Latin square design was employed, with a row length of 2.5 m, plant spacing of 25 cm, and 14 plants per row, with two replications. The four subpopulations were also planted in Menghang Town, Jinghong City, Xishuangbanna Dai Autonomous Prefecture, during the winter of 2021, using a randomized complete block design with a row length of 2.5 m, 14 plants per row, and plant spacing of 25 cm, with two replications.

### 4.2. Phenotyping of Flowering Time-Related Traits

The days to anthesis (DTA) for the RILs of the multiparent population was assessed by calculating the number of days from plant emergence to the stage when more than 50% of the plant tassels developed. Similarly, the days to silking (DTS) of the RILs was evaluated by measuring the number of days from plant emergence to the stage when more than 50% of the plants began to exhibit silking. The anthesis and silking interval (ASI) of the RILs was calculated as the number of days between the DTA and DTS.

### 4.3. Statistical Analyses of the Phenotypic Data

Phenotypic data were analyzed using R software, involving descriptive statistical analysis and correlation analysis, and corresponding graphs were created to visualize the data. The SAS9.1 MIXED model was employed to fit the multi-environmental phenotypic data of the maize multiparent population, enabling the calculation of the Best Linear Unbiased Prediction (BLUP). BLUP values were subsequently used for GWAS analysis.

### 4.4. Genotyping-by-Sequencing (GBS)

Genomic DNA was extracted from the seedling leaves of each RIL using the cetyltrimethylammonium bromide (CTAB) method [66]. Subsequently, the genomic DNA was digested with the restriction endonucleases *PstI* and *MspI* (New England BioLabs, Ipswich, MA, USA) and ligated to barcode adapters using T4 ligase (New England BioLabs). GBS DNA libraries were constructed and sequenced according to the GBS protocol [67].

Following ligation, all samples were pooled and purified using a QIAquick PCR Purification Kit (QIAGEN, Valencia, CA, USA). Polymerase chain reaction (PCR) was performed using primers complementary to both the adapters. Subsequently, PCR products were purified and quantified using the Qubit dsDNA HS Assay Kit (Life Technologies, Grand Island, NY, USA). Following the selection of PCR products with a size range of 200–300 bp using an Egel system (Life Technologies), the concentration of the library was estimated using a Qubit 2.0 Fluorometer and a Qubit dsDNA HS Assay Kit (Life Technologies). Sequencing was performed on an Ion Proton sequencer (Life Technologies, software version 5.10.1) using the P1v3 chips. Subsequently, sequencing reads were generated using TASSEL v5.0 [68]. Prior to the TASSEL analysis, 80 poly(A) bases were appended to the 3′ ends of all sequencing reads. For comparative analysis, the B73_V4 reference genome sequence was used, and the analysis was performed using the Sentieon software (parameter “bwa mem -k 32-M-R”) [69]. Samtools (using the parameter rmdup) was used to sort and de-weigh the comparison results. To obtain high-quality SNPs, the following filtering criteria were applied: (1) heterozygous SNPs were filtered out, and only homozygous SNPs were retained; (2) SNPs with a missing rate exceeding 20% were removed, ensuring that each SNP was present in at least 80% of the samples; and (3) SNPs with a minimum allele frequency (MAF) < 0.05 were removed. Finally, a total of 588,416 high-quality SNPs were generated and annotated using the ANNOVAR software tool (v2013-05-20) [70].

### 4.5. Population Structure and Linkage Disequilibrium Analysis

The Admixture [71] software was used to analyze the population structure using genotypic data derived from GBS. Considering that the experimental population comprised four subpopulations, the optimal number of clusters (K) was determined at K = 4. The R package was used to visualize the population structure. Pop LD decay (https://github.com/BGI-shenzhen/PopLDdecay, accessed on 13 April 2023) [72] software was used to assess the degree of linkage disequilibrium (r^2^) between the two markers, and the LD decay plot was generated using Plot_OnePop.pl software. 

### 4.6. Genome-Wide Association Analysis

We used GEMMA (http://www.xzlab.org/software.html, accessed on 29 May 2023) to identify the association between SNPs and flowering-related traits using a mixed linear model (MLM). In this model, population structure was considered a fixed effect, whereas kinship among individuals was incorporated as a random effect to account for the effects of both population structure and kinship.

The model is represented as follows:y = Xα + Zβ + Wμ + e(1)
where y represents the phenotype, X is a fixed-effect indicator matrix, α is an estimated fixed-effect parameter, Z is an indicator matrix, β is the effect of the SNP, W is a random-effect matrix, µ is a predicted random individual effect, and e is a vector of random residual effects following e~ (0, δ_e_^2^).

Association analysis of the three flowering-related traits (DTA, DTS, and ASI) in maize was conducted in the multiparent population. The significance level was calculated using the formula −log10 (1/590,816), where 588,416 represents the number of SNPs used in the experiment. A significance threshold of −log10(*p*) = 5 was applied to identify potential candidate SNPs [73].

### 4.7. Candidate Gene Identification

Based on GWAS analysis, SNPs that were significantly associated with maize flowering traits (DTA, DTS, and ASI) were identified. Sequences flanking each SNP showing significant associations were extracted and aligned with the B73 _V4 reference sequence. Subsequently, the BLAST function in MaizeGDB was used to locate SNP positions. Candidate genes were then screened within a 20 kb range upstream and downstream of the SNPs to identify potential candidate genes associated with the significant SNPs. Simultaneously, positive BAC libraries encompassing QTLs for flowering traits (DTA, DTS, and ASI) were screened to acquire genomic sequences at corresponding positions. Subsequently, the candidate genes associated with maize flowering traits (DTA, DTS, and ASI) were validated by comparison with the B73_V4 reference. The selected candidate genes were functionally annotated and predicted using the MaizeGDB and NCBI (https://www.ncbi.nlm.nih.gov/) databases.

### 4.8. Haplotype Analysis

A region extending 1Mb upstream and downstream of the genes *Zm00001d028325*, *Zm00001d028615*, and *Zm00001d048680* was selected, and the block value was calculated using the Haploview software and displayed on a heat map. Haplotypes within these genic regions were analyzed based on the positions of the *Zm00001d028325*, *Zm00001d028615*, and *Zm00001d048680* genes. Box plots were then generated, illustrating the relationship between the haplotypes and phenotypes. 

## Figures and Tables

**Figure 1 plants-13-01032-f001:**
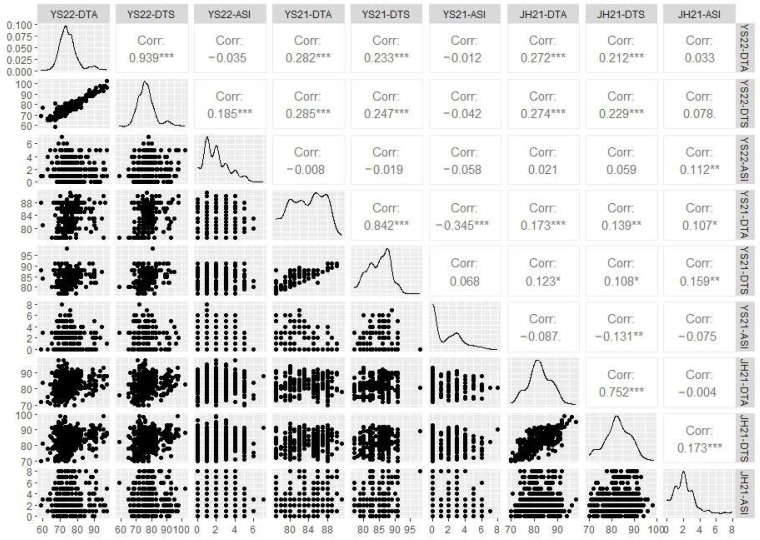
Correlation analysis of maize flowering time-related traits. YS22 represents the trial conducted in Yanshan in 2022; YS21 represents the trial conducted in Yanshan in 2021; JH21 represents the trial conducted in Jinghong in 2021. DTA, days to anthesis; DTS, days to silking; ASI, anthesis and silking interval. * means *p* ≤ 0.05, ** means *p* ≤ 0.01, *** means *p* ≤ 0.001.

**Figure 2 plants-13-01032-f002:**
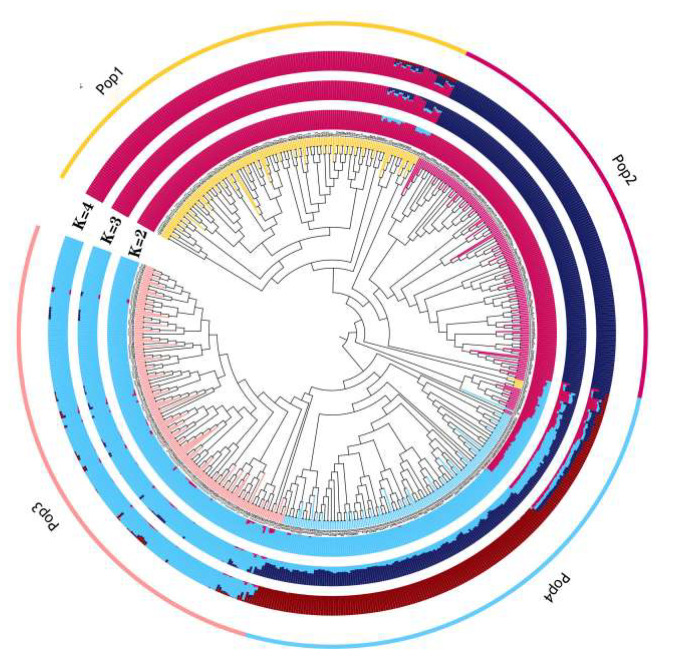
Analysis of population structure of the multiparent population. Diagram of the genetic structure of the population from the inner circle outwards for K = 2, 3, 4. Pop1 is yellow, Pop2 is red, Pop3 is pink and Pop is blue.

**Figure 3 plants-13-01032-f003:**
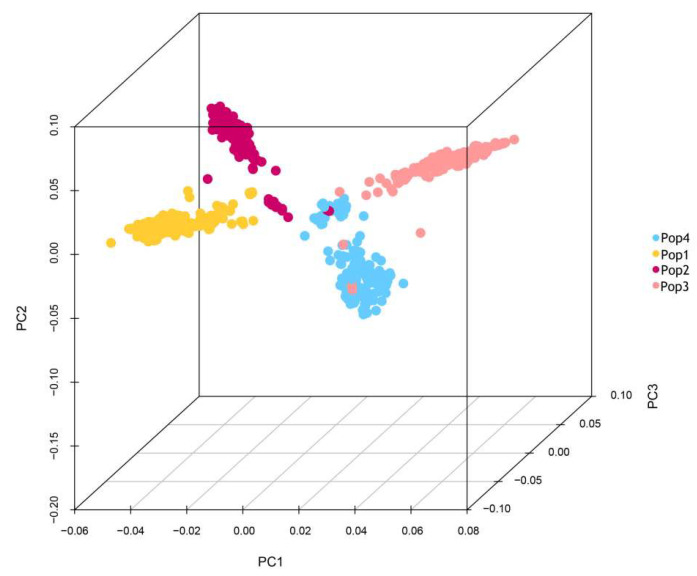
Three-dimensional diagram of principal component analysis. The four subpopulations, Pop1, Pop2, Pop3, and Pop4, are clearly clustered.

**Figure 4 plants-13-01032-f004:**
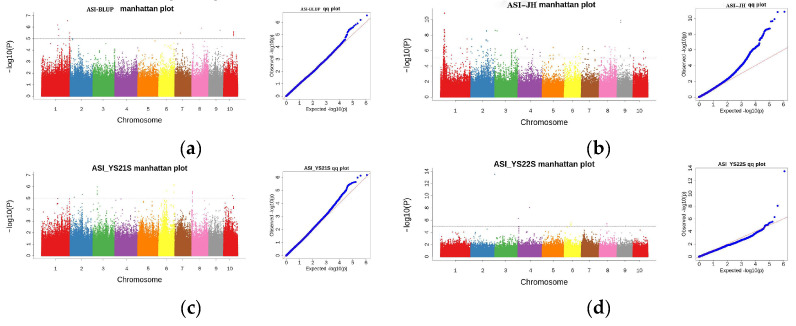
The Manhattan and Q-Q plots depict the SNPs associated with ASI in the multiparent population across multiple environments. (**a**) Manhattan plot and quantile–quantile plot for BLUP value for ASI; (**b**) Manhattan plot and quantile–quantile plot for ASI in 2021 in Jinghong; (**c**) Manhattan plot and quantile–quantile plot for ASI in 2021 in Yanshan; (**d**) Manhattan plot and quantile–quantile plot for ASI in 2022 in Yanshan.

**Figure 5 plants-13-01032-f005:**
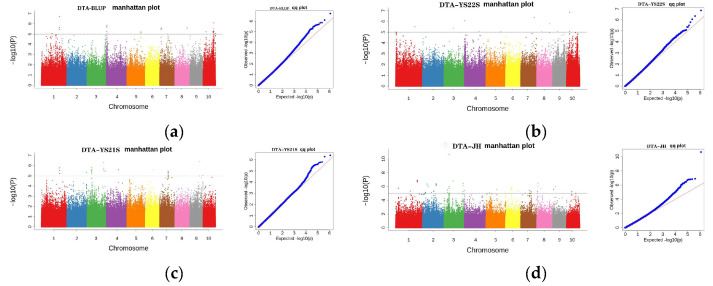
Manhattan and Q-Q plots depict the SNPs associated with DTA in the multiparent population across multiple environments. (**a**) Manhattan plot and quantile–quantile plot for BLUP value for DTA; (**c**) Manhattan plot and quantile–quantile plot for DTA in 2021 in Yanshan; (**b**) Manhattan plot and quantile–quantile plot for DTA in 2022 in Yanshan; (**d**) Manhattan plot and quantile–quantile plot for DTA in 2021 in Jinghong.

**Figure 6 plants-13-01032-f006:**
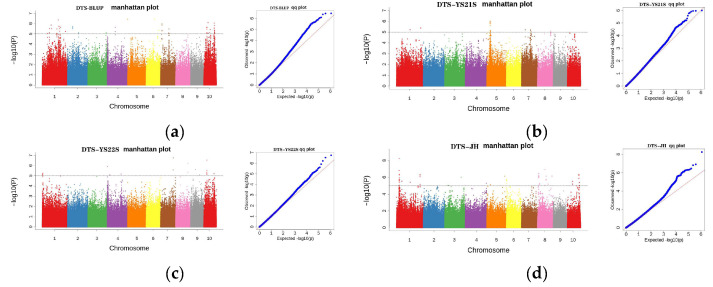
Manhattan and Q-Q plots depict the SNPs associated with DTS in the multiparent population across multiple environments. (**a**) Manhattan plot and quantile–quantile plot for BLUP value for DTS; (**b**) Manhattan plot and quantile–quantile plot for DTS in 2021 in Yanshan; (**c**) Manhattan plot and quantile–quantile plot for DTS in 2022 in Yanshan; (**d**) Manhattan plot and quantile–quantile plot for DTS in 2021 in Jinghong.

**Figure 7 plants-13-01032-f007:**
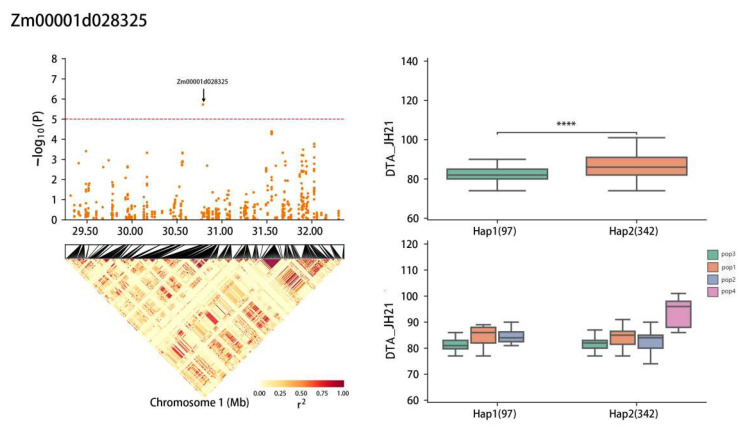
Haplotype analysis of *Zm00001d028325* (JH21-DTA). **** means *p* ≤ 0.0001.

**Figure 8 plants-13-01032-f008:**
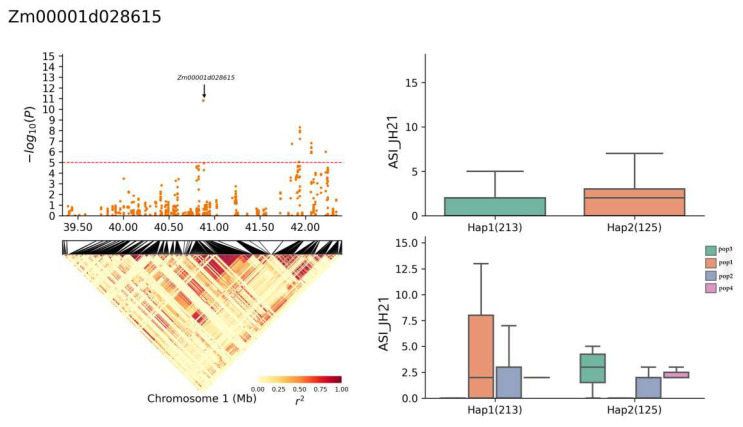
Haplotype analysis of *Zm00001d028615* (JH21-ASI).

**Figure 9 plants-13-01032-f009:**
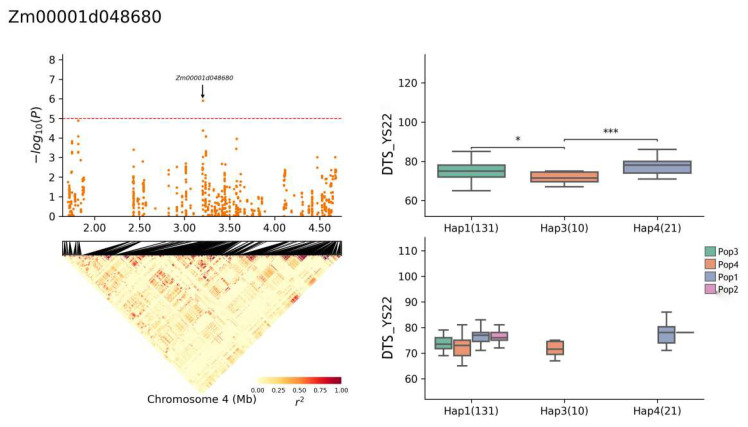
Haplotype analysis of *Zm00001d048680* (YS22-DTS). * means *p* ≤ 0.05, *** means *p* ≤ 0.001.

**Figure 10 plants-13-01032-f010:**
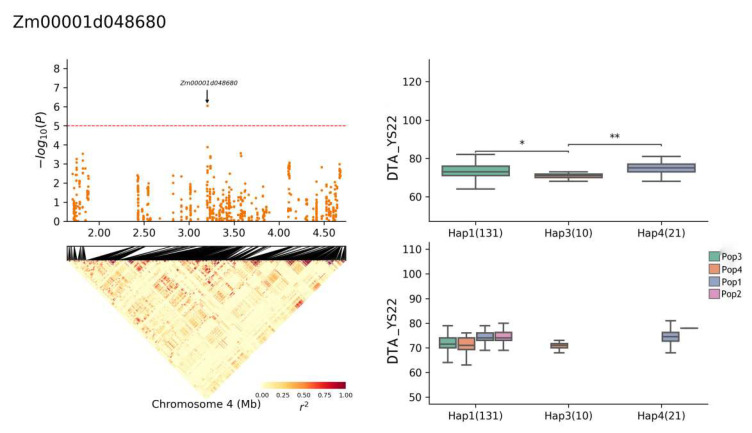
Haplotype analysis of *Zm00001d048680* (YS22-DTA). * means *p* ≤ 0.05, ** means *p* ≤ 0.01.

**Figure 11 plants-13-01032-f011:**
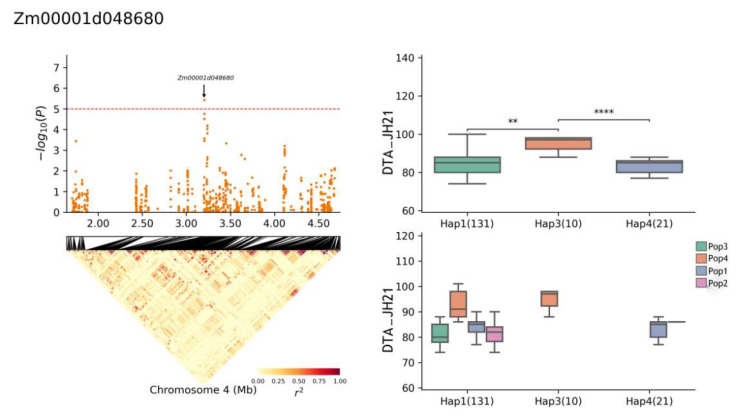
Haplotype analysis of *Zm00001d048680* (JH21-DTA).** means *p* ≤ 0.01, **** means *p* ≤ 0.0001.

**Table 1 plants-13-01032-t001:** Descriptive statistical analysis of maize flowering time-related traits across different environments.

Environment/Traits	Max	Mix	Average Value	Standard Deviation	Variance	Coefficient of Variation
YS22-DTA	59	97	75.03	5.774	33.337	0.076955884
YS22-DTS	59	102	76.87	5.885	34.635	0.076557825
YS22-ASI	0	7	1.9316	1.33785	1.79	0.692612342
YS21-DTA	74	101	85.41	5.919	35.039	0.069301019
YS21-DTS	77	101	86.07	5.197	27.007	0.060381085
YS21-ASI	0	15	1.83	2.554	6.524	1.395628415
JH21-DTA	70	98	82.37	4.955	24.553	0.060155396
JH21-DTS	70	99	82.72	5.468	29.9	0.066102515
JH21-ASI	0	8	1.986	1.28866	1.661	0.648872105

**Table 2 plants-13-01032-t002:** Broad-sense heritability of flowering time-related traits in maize.

Traits	Genetic Variance (Vg)	Residual Variance (Ve)	Number of Environments (L)	Heritability H^2^ = Vg/(Vg + Ve/L)
DTA	5.27	25.57	3	0.382068632
DTS	6.296	25.932	3	0.421419009
ASI	0.209	4.6203	3	0.119490023

**Table 3 plants-13-01032-t003:** List of candidate genes identified through GWAS in the multiparent population for the flowering time-related traits in maize.

Gene ID	Chr.	Environments/Traits	Physical Position	Functional Annotation
*Zm00001d020229*	7	YS21-DTA, YS21-DTS, BULP-DTS	101,183,127–101,183,962	Reticulon-like protein B1
*Zm00001d021470*	7	YS22-DTA, YS22-DTS	153,075,349–153,079,559	Glutathione S-transferase F10
*Zm00001d028295*	1	JH21-DTS, JH21-ASI	29,497,242–29,504,081	Protein LOW PSII ACCUMULATION 3, chloroplastic
*Zm00001d028325*	1	JH21-DTS, JH21-DTA	30,793,737–30,799,897	Cytochrome P450 90B1
*Zm00001d028615*	1	JH21-ASI	40,886,065–40,889,048	Probable protein phosphatase 2C 31
*Zm00001d033665*	1	JH21-DTS, JH21-ASI	269,986,885–269,988,386	30S ribosomal protein S13, chloroplastic
*Zm00001d033666*	1	JH21-DTS, JH21-ASI	269,989,007–269,992,773	Probable plastid–lipid-associated protein 4, chloroplastic
*Zm00001d039455*	3	JH21-ASI	4,833,807–4,837,335	PRA1 family protein F3
*Zm00001d039456*	3	JH21-ASI	4,838,566–4,841,548	
*Zm00001d047209*	9	JH21-ASI	122112583–122121985	Calmodulin-binding transcription activator 1
*Zm00001d051680*	4	YS22-DTA, YS22-DTS	166,593,637–166,595,477	Ferredoxin–NADP reductase, chloroplastic
*Zm00001d048680*	4	YS22-DTA, YS22-DTS, JH21-DTA, BLUP-DTA	3,197,914–3,204,309	Cysteine-rich receptor-like protein kinase 10

**Table 4 plants-13-01032-t004:** Important haplotypes associated with flowering-related traits.

Gene ID	Position	Haplotype	Hap_Sample_Num ^1^
Zm00001d028325	Chr1: 29,316,772–32,297,235	CCACA(Hap1)	97
		TTGTC(Hap2)	342
		CCGCA(Hap3)	3
Zm00001d028615	Chr1: 39,389,512–42,337,172	TCGCAAGGG(Hap1)	213
		GGTACGATA(Hap2)	125
Zm00001d048680	Chr4: 1,703,082–4,679,821	CTAGTATGATCGCTA(Hap1)	131
		TCATTGAGATAACAC(Hap2)	3
		CCATTGTGGTCGCAC(Hap3)	10
		TCGTTGTAATCACAC(Hap4)	21

^1^ hap_sample_num is the total number of identical haplotypes.

**Table 5 plants-13-01032-t005:** Important haplotype phenotypic data and phenotypic variance explained (PVE) by the haplotypes associated with flowering traits.

Gene ID	Environments/Traits	Average	GWAS-PVE
Zm00001d028325	JH21-DTA	82.5 (Hap1)	5.90%
		87.0 (Hap2)	
Zm00001d028615	JH21-ASI	2.1 (Hap1)	12.43%
		1.9 (Hap2)	
Zm00001d048680	YS22-DTS	75.8 (Hap1)	4.67%
		71.7 (Hap3)	
		77.5 (Hap4)	
	YS22-DTA	74.1 (Hap1)	5.06%
		69.7 (Hap3)	
		74.8 (Hap4)	
	JH21-DTA	85.5 (Hap1)	3.82%
		94.8 (Hap3)	
		83.9 (Hap4)	

**Table 6 plants-13-01032-t006:** Positive and negative EBVs of different traits in four subpopulations.

Subpopulation	EBV ^1^ (DTA ^2^)		EBV (DTS ^2^)		EBV (ASI ^2^)	
	+	−	+	−	+	−
Pop3	63	126	79	110	67	122
Pop4	87	64	92	59	85	66
Pop1	87	88	76	99	69	106
Pop2	84	79	72	88	65	95

^1^ EBV: Estimated breeding values; + and − represent positive and negative EBVs. ^2^ DTA = day to anthesis, DTS = day to silking, ASI = anthesis and silking interval.

**Table 7 plants-13-01032-t007:** The average ASI days of each population in different environments.

Environment	Population	Average Day
JH21	pop1	2.67
	pop2	2.17
	pop3	2.45
	pop4	3.85
YS21	pop1	1.94
	pop2	1.77
	pop3	1.33
	pop4	2.98
YS22	pop1	2.15
	pop2	1.93
	pop3	2.03
	pop4	2.13

**Table 8 plants-13-01032-t008:** Parental lines used for the development of multiparent populations.

Line	Pedigree	Heterotic Group	Ecotype
CML384	P502-C1-771-2-2-1-3-B	Reid	Tropical
CML444	P43-C9-1-1-1-1-1	nonReid	Tropical
CML171	Pool25QPM	nonReid	Tropical
Shen137	Derived from US hybrid (6JK111)	nonReid	Temperate
Ye107	Form American	Reid	Temperate

## Data Availability

Data are contained within the article and Appendix A.

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
