# Peer review of "Studies on Candidate Genes Related to Flowering Time in a Multiparent Population of Maize Derived from Tropical and Temperate Germplasm"

_plants, 2024, doi:10.3390/plants13071032_

Round 1

Reviewer 1 Report

Comments and Suggestions for Authors

The article "Studies on Candidate Genes Related to Flowering Time in a Multiparent Population of Maize Derived From Tropical and Temperate Germplasm" provides an in-depth genetic analysis of maize flowering time. Here are more specific and detailed opinions on various aspects of the study.

he use of a multiparent population, combining both tropical and temperate maize germplasm, is a strength, as it likely captures a broad genetic diversity and allows for the detection of alleles relevant under different climatic conditions.

The identification of 556 SNPs and 12 candidate genes significantly associated with flowering time-related traits indicates a comprehensive analysis. However, the functional roles of these candidate genes in the regulation of flowering time could be better detailed to enhance understanding of their biological significance.

While the study is well-executed, the connection between the identified genetic markers and their functional impact on flowering time could be further explored through functional genomics studies, such as gene expression analysis and knockout experiments, to validate the roles of these candidate genes.

Further functional validation and exploration of gene-environment interactions would enhance the understanding of the complex nature of flowering time regulation in maize.

Reviewer 2 Report

Comments and Suggestions for Authors

I would think that the data presented here are overall fine, but I have a few comments concerning the data as below.

1. The authors have presented only statistics of phenotypic data. More detailed phenotypic data such as a frequency distribution of each trait would be helpful for readers to understand the trait characteristics.

2. The authors showed a phylogenetic tree of RILs. I do not think that a phylogenetic analysis is popular for RILs (artificially made populations). I would recommend that the removal of this data.
